# Estimated glomerular filtration rate predicts 30-day mortality in medical emergency departments: Results of a prospective multi-national observational study

**Laurent Haas**[1]\*, **Andreas Eckart**[1], **Sebastian Haubitz**[2], **Beat Mueller**[2], **Philipp Schuetz**[2], **Stephan Segerer**[1]

**1** Division of Nephrology, Dialysis and Transplantation, University Department of Medicine, Kantonsspital Aarau, Aarau, Switzerland, **2** Division of General Internal and Emergency Medicine, University Department of Medicine, Kantonsspital Aarau, Aarau, Switzerland

\* haaslaurent1985@gmail.com

## Abstract

### Background

Renal failure is common in patients seeking help in medical emergency departments. Decreased renal function is associated with increased mortality in patients with heart failure or sepsis. In this study, the association between renal function (reflected by estimated glomerular filtration rate (eGFR) at the time of admission) and clinical outcome was evaluated.

### Methods/Objectives

Data was used from a prospective, multi-national, observational cohort of patients treated in three medical emergency departments of tertiary care centers. The eGFR was calculated from the creatinine at the time of admission (using the **C**hronic **K**idney **D**isease-**Epi**demiology Collaboration equation,**CKD-EPI**). Uni- and multivariate regression models were used for eGFR and 30-day mortality, in hospital mortality, length of stay and intensive care unit admission rate.

### Results

6983 patients were included. The 30-day mortality was 1.8%, 3.5%, 6.9%, 11.1%, 13.6%, and 14.2% in patients with eGFR of above 90, 60–89, 45–59, 30–44, 15–29, and <15 ml/min/1.73m$^2$, respectively. Using multivariate regression, the adjusted odds ratio (OR) was 2.31 (for 15–29 ml/min/1.73m$^2$, 95% confidence interval 1.36 to 3.90, p = 0.002) and 3.73 (for eGFR <15ml/min/1.73m$^2$ as compared to >90 ml/min/1.73m$^2$, 95% CI 2.04 to 6.84, p<0.001). For 10 ml/min/1.73m$^2$ decrease in eGFR the OR for the 30-day mortality was 1.15 (95% CI1.09 to 1.22, p<0.001).The eGFR was also significantly associated with in-hospital mortality, the percentage of ICU-admissions, and with a longer hospital stay. No association was found with hospital readmission within 30 days. As limitations, only eGFR at admission was available and the number of patients on hemodialysis was unknown.

**Data Availability Statement:** Extra data can be accessed via the Dryad data repository at http://

datadryad.org/ with the DOI: 10.5061/dryad.
71638rk.

**Funding:** The work was supported by a grant of the
Fundação Pesquisa e Desenvolvimento
Humanitario. The TRIAGE Project was supported in
part by the Swiss National Science Foundation
(SNSF Professorship, PP00 P3_150531/1), the
Swiss Academy for Medical Sciences
(Schweizerische Akademie der Medizinischen
Wissenschaften [SAMW]), and the Research
Council of the Kantonsspital Aarau (1410.000.044).
The funders had no role in study design, data
collection and analysis, decision to publish, or
preparation of the manuscript.

**Competing interests:** The authors have declared
that no competing interests exist.

## Conclusion

Reduced eGFR at the time of admission is a strong and independent predictor for adverse
outcome in this large population of patients admitted to medical emergency departments.

## Introduction

Impaired renal function represents a diagnostic and therapeutic challenge for the emergency
department [1]. Independent of the cause of admission, renal insufficiency might impact the
disease courses in several ways. It potentially adds disturbances to volume status, water, acid
base, and electrolytes as clinical challenges. Renal failure leads to a proinflammatory but also
immunocompromised state, which promotes injury to distal organs including lung, heart and
nervous system [2–5]. Kidney function might influence the interpretation of diagnostic tests
and biomarkers (e.g troponins and natriuretic peptides) [6]. Finally, dosing of various drugs
has to be adjusted, which entails the danger of both overdosing resulting in toxicity, and
underdosing with insufficient therapeutic effect [7, 8]

   Decreased renal function in both acute kidney injury, as well as chronic kidney disease has
been shown to be associated with increased in-hospital mortality, increased resource utiliza-
tion, and increased hospital cost of care [9–15]. For example the outcome in patients with sep-
sis or heart failure is significantly worse in those with kidney disease [9, 13, 16–20].

   We hypothesize that impaired renal function, reflected by eGFR (CKD-EPI) from admis-
sion creatinine at time of medical emergency department (ED) presentation, might be a
marker of poor outcome. We used data from a large, multi-national, prospective, observational
study, initially designed to investigate triage prioritization improvement by the addition of bio-
markers [21], to evaluate associations with clinical outcome.

## Methods

### Study design

We used data of the TRIAGE study. The purpose of this study was to evaluatewhether the
addition of biomarkers would improve early risk stratification in a "real life" population of
medical patients presenting to the ED of tertiarycare hospitals. Hier meinen Comment zu
Rev2 Frage 3 einfügen?. A total of 7132 consecutive patients were recruited from March 2013
to October 2014 [21, 22]. After excluding patients with missing serum creatinine values, a
cohort of 6,983 patients was included in the current investigation (4,554 from Aarau (Switzer-
land), 1,460 from Paris (France), and 969 from and Clearwater (FL, USA). The baseline char-
acteristics per center are included in the supplemental data (S1 Table).

   As an observational, quality control study, the Institutional Review Boards of the three hos-
pitals approved the study and waived the need for individual informed consent (Ethikkommis-
sion Kanton Aargau (EK 2012/059); CCTIRS-Comité consultatif sur le traitement de
l'information en matière de recherche; CPP ID RCB:2013-A00129-36) MPM-SAH Institu-
tional ReviewBoard, Clearwater; FL,IRB number 2013_005). The study was registered at "Clin-
icalTrials.gov" registration website and the study protocol has been published [21].

### Data collection and definitions of diagnoses

Laboratory evaluation was only included from the time of ED admission and was part of the
routine workup. All participants provided a medical history and underwent a physical

examination. Congestive heart failure, coronary heart disease, hypertension, chronic obstructive pulmonary disease, dementia, diabetes mellitus, cancer, renal failure and history of stroke were documented as co-morbidities. Based on the information at discharge from the ED, patients were grouped into main diagnosis groups (e.g. infectious, cardiovascular) by two independent physicians. Management throughout the hospital stay was at the discretion of the treating physician, independent of the research team. A standardized telephone interview was performed 30 days after hospital admission to assess vital status, functional outcome and hospital readmissions.

### Primary and secondary endpoints

The primary endpoint was all-cause 30-day mortality. Secondary endpoints were in-hospital mortality, admission to the intensive care unit (ICU), hospital readmission within 30 days, and length of hospital stay (LOS). The decision for ICU admission was left to the discretion of the treating physicians.

### Assessment of renal function

Serum creatinine was measured by the clinical chemistry laboratory photometrically using the Jaffe reaction (Siemens, Dimension Vista® System, Flex ® reagent cartridge CREA). The eGFR was calculated according to the CKD-EPI 2009 equation expressed for specified sex and serum creatinine level as follows [23]:

$$GFR = 141 \times \min (S_{cr}/\kappa, 1)^{\alpha} \times \max(S_{cr}/\kappa, 1)^{-1.209} \times 0.993^{Age} \times 1.018 \text{ [if female]}.$$

$S_{cr}$ is serum creatinine in μmol/L, κ is 61.9 for females and 79.6 for males, α is -0.329 for females and -0.411 for males, min indicates the minimum of $S_{cr}/\kappa$ or 1, and max indicates the maximum of $S_{cr}/\kappa$ or 1. No adjustment was performed for black or African ethnicity as the information was not available from the data collection.

The eGFR was categorized into six categories (eGFR >90 ml/min/1.73 m², 60–89 ml/min/1.73 m², 45–59 ml/min/1.73 m², 30–44 ml/min/1.73 m²,15–29 ml/min/1.73 m², <15 ml/min/1.73 m²) as proposed by the Kidney Disease Improving Global Outcomes (KIDIGO) Guidelines 2012 for chronic kidney diseases [24].

### Statistical analysis

Descriptive statistics, including mean with standard deviation (SD), median with interquartile range (IQR), and frequencies were used to describe the study population, when appropriate. To evaluate associations of eGFR with primary and secondary endpoints univariate and multivariate logistic regression analyses were applied and odds ratios (OR) with 95% confidence intervals (CI) were reported. The models were adjusted in a stepwise algorithm for age, gender, main diagnosis and comorbidities. The discriminatory power was described using area under the receiver operating characteristics curve analysis (AUC). All tests were two-tailed and carried out at 5% significance levels (exceptions stated separately). Analyses were performed with Stata 12.1 (Stata Corp., College Station, TX, USA).

## Results

### Patient characteristics

A total of 6,983 patients with follow-up data were included in the analysis. The median age was 62 years and 46.7% of patients were female (**Table 1**). The three most frequent diagnoses were cardiovascular diseases (23.4%), neurological diseases (22.2%) and infections (14.7%). Patients had a high burden of comorbidities including hypertension (39.7%), coronary heart disease

**Table 1. Baseline characteristics of the total cohort and stratified by eGFR categories.**

| | Total cohort | eGFR (ml/min/1.73m$^2$) | | | | | |
| --- | --- | --- | --- | --- | --- | --- | --- |
| | | >90 | 60–89 | 45–59 | 30–44 | 15–29 | <15 |
| Number (%) | 6983 | 2544 (36%) | 2504 (35%) | 889 (12%) | 561 (9%) | 309 (5%) | 176 (3%) |
| Age(years; median,(IQR) | 62 (46, 76) | 44 (31, 56) | 67 (55, 76) | 77 (68, 84) | 79 (69, 85) | 79 (71, 85) | 69 (58, 82) |
| Male Sex | 3723 (53.3%) | 1288 (50.6%) | 1403 (56.0%) | 463 (52.1%) | 308 (54.9%) | 162 (52.4%) | 99 (56.2%) |
| **Vital signs, median (IQR)** | | | | | | | |
| Systolic blood pressure (mmHg) | 137 (121, 154) | 133 (120, 148) | 142 (126, 158) | 141 (123, 160) | 137.50 (117, 155) | 127 (105, 148) | 130 (107, 155) |
| Diastolic blood pressure (mmHg) | 80 (70, 90) | 82 (73, 91) | 81 (72, 92) | 78 (68, 90) | 75 (65, 85) | 69 (58, 81) | 70 (56, 85) |
| Pulse rate (bpm) | 83 (71, 97) | 84 (73, 97) | 82 (70, 96) | 82 (70, 93) | 82 (71, 99) | 83 (70, 98) | 83 (69, 98) |
| O2 Saturation (%) | 96 (94, 98) | 97 (96, 99) | 96 (94, 98) | 96 (93, 97) | 95 (92, 97) | 95 (93, 97.4) | 96 (92, 98) |
| Temperature (˚C) | 36.8 (36.4, 37.2) | 36.8 (36.5, 37.2) | 36.8 (36.4, 37.2) | 36.8 (36.4, 37.3) | 36.8 (36.5, 37.5) | 36.7 (36.3, 37.3) | 36.7 (36.2, 37.2) |
| **Laboratory results, median (IQR)** | | | | | | | |
| Hemoglobin (g/l), | 13.6 (12.1, 14.8) | 14 (12.8, 15.1) | 13.8 (12.5, 14.9) | 13.1 (11.6, 14.5) | 12.3 (10.8, 13.8) | 11.4 (9.8, 13.0) | 10.9 (9.4, 12.4) |
| Leukocyte count (G/l) | 8.37 (6.57, 10.96) | 8.05 (6.36, 10.4) | 8.32 (6.50, 10.71) | 8.59 (6.80, 11.24) | 9.29 (6.90, 12.59) | 9.30 (6.84, 13.01) | 9.26 (7.09, 13.20) |
| Glucose (mmol/l) | 6.1 (5.4, 7.5) | 5.7 (5.1, 6.5) | 6.3 (5.5, 7.5) | 6.6 (5.7, 8.2) | 7 (5.8, 9.3) | 7.1 (6.0, 9.2) | 6.9 (5.3, 8.9) |
| Creatinine (μmol/l) | 81 (67, 103) | 64 (56, 74) | 82 (72, 94) | 106 (91, 117) | 136 (112, 156) | 209 (179, 243) | 487 (369, 686) |
| eGFR (ml/min/1.73m2) | 80 (57, 98) | 104 (97, 115) | 76 (68, 83) | 53 (49, 56) | 38 (34, 41) | 23 (18, 26) | 8 (6, 12) |
| CRP (mg/l) | 5.4 (<3, 27.8) | 1 (<3, 11.7) | 4.8 (<3, 22.5) | 9.3 (<3, 48.7) | 18.1 (4.9, 74.7) | 32.9 (7.4, 116) | 32.3 (6.1, 159) |
| PCT (μg/l) | 0.08 (0.06, 0.13) | 0.07 (0.05, 0.10) | 0.07 (0.06, 0.11) | 0.09 (0.07, 0.14) | 0.12 (0.08, 0.25) | 0.21 (0.12, 0.51) | 0.47 (0.21, 1.64) |
| Copeptin (pmol/l) | 10.8 (4.5, 39.2) | 5.8 (3.3, 12.5) | 10.1 (4.7, 31.6) | 22.6 (8.6, 61.0) | 42.9 (17.5, 101.0) | 85.6 (41.6, 161.1) | 136 (59.7, 218.0) |
| **Main diagnosis, number, (%)** | | | | | | | |
| Infection | 1026 (14.7%) | 333 (13.1%) | 323 (12.9%) | 145 (16.3%) | 106 (18.9%) | 81 (26.2%) | 38 (21.6%) |
| Cardiovascular | 1636 (23.4%) | 512 (20.1%) | 648 (25.9%) | 226 (25.4%) | 148 (26.4%) | 62 (20.1%) | 40 (22.7%) |
| Metabolic | 190 (2.7%) | 58 (2.3%) | 44 (1.8%) | 16 (1.8%) | 24 (4.3%) | 17 (5.5%) | 31 (17.6%) |
| Cancer | 339 (4.9%) | 101 (4.0%) | 126 (5.0%) | 42 (4.7%) | 45 (8.0%) | 15 (4.9%) | 10 (5.7%) |
| Neurological | 1549 (22.2%) | 575 (22.6%) | 613 (24.5%) | 210 (23.6%) | 96 (17.1%) | 42 (13.6%) | 13 (7.4%) |
| Gastrointestinal | 977 (14.0%) | 426 (16.7%) | 301 (12.0%) | 106 (11.9%) | 67 (11.9%) | 48 (15.5%) | 29 (16.5%) |
| Pulmonary | 292 (4.2%) | 115 (4.5%) | 104 (4.2%) | 41 (4.6%) | 20 (3.6%) | 9 (2.9%) | 3 (1.7%) |
| Other | 974 (13.9%) | 424 (16.7%) | 345 (13.8%) | 103 (11.6%) | 55 (9.8%) | 35 (11.3%) | 12 (6.8%) |
| **Comorbidities, number, (%)** | | | | | | | |
| Cancer | 960 (13.7%) | 251 (9.9%) | 377 (15.1%) | 145 (16.3%) | 102 (18.2%) | 57 (18.4%) | 28 (15.9%) |
| Renal failure | 864 (12.4%) | 30 (1.2%) | 56 (2.2%) | 174 (19.6%) | 264 (47.1%) | 219 (70.9%) | 121 (68.8%) |
| Congestive heart disease | 481 (6.9%) | 38 (1.5%) | 133 (5.3%) | 124 (13.9%) | 98 (17.5%) | 63 (20.4%) | 25 (14.2%) |
| COPD | 355 (5.1%) | 95 (3.7%) | 127 (5.1%) | 60 (6.7%) | 43 (7.7%) | 25 (8.1%) | 5 (2.8%) |
| Coronary heart disease | 834 (11.9%) | 146 (5.7%) | 350 (14.0%) | 153 (17.2%) | 101 (18.0%) | 55 (17.8%) | 29 (16.5%) |
| Dementia | 220 (3.2%) | 12 (0.5%) | 68 (2.7%) | 76 (8.5%) | 39 (7.0%) | 14 (4.5%) | 11 (6.2%) |
| Diabetes mellitus | 1075 (15.4%) | 183 (7.2%) | 374 (14.9%) | 191 (21.5%) | 169 (30.1%) | 96 (31.1%) | 62 (35.2%) |
| History of Stroke | 564 (8.1%) | 124 (4.9%) | 235 (9.4%) | 118 (13.3%) | 53 (9.4%) | 30 (9.7%) | 4 (2.3%) |
| Hypertension | 2769 (39.7%) | 486 (19.1%) | 1138 (45.4%) | 551 (62.0%) | 335 (59.7%) | 182 (58.9%) | 77 (43.8%) |

(11.9%), diabetes (15.4%), and cancer (13.7%). Most patients were treated as inpatients (72.8%).

## Associations of eGFR with primary and secondary endpoint

A total of 325 Patients (4.7%) died within thirty days of admission. The percentage was 1.8% in patients with an eGFR >90 ml/min/1.73m$^2$ compared to 14.2% in patients with an eGFR <15ml/min/1.73m$^2$ (**Table 2**).

Primary and secondary outcomes stratified by eGFR groups and with eGFR as a continuous variable are illustrated in **Tables 2 and 3**. In the unadjusted model for 30-day mortality the OR was 1.94 (95% CI 1.35 to 2.77, p<0.001) in the eGFR group of 60–89 ml/min/1.73 m$^2$ and with

**Table 2. Frequencies of primary and secondary endpoints according to eGFR groups and associations of eGFR stratified by groups with adverse clinical outcome in univariate and multivariate regression analyses.**

| | Total cohort | eGFR (ml/min/1.73m$^2$) | | | | | |
| --- | --- | --- | --- | --- | --- | --- | --- |
| | | >90 | 60–89 | 45–59 | 30–44 | 15–29 | <15 |
| **30-day mortality, number (%)** | 325/6983 (4.65%) | 47/2544 (1.8%) | 88/2504 (3.5%) | 61/889 (6.9%) | 62/561 (11.1%) | 42/309 (13.6%) | 25/176 (14.2%) |
| Odds ratio (95% CI), p-value | | | | | | | |
| Unadjusted model | | Ref | 1.94 (1.35 to 2.77), p<0.001 | 3.91 (2.65 to 5.77), p<0.001 | 6.60 (4.46 to 9.76), p<0.001 | 8.36 (5.41 to 12.91), p<0.001 | 8.80 (5.27 to 14.68), p<0.001 |
| Model 1 | | Ref | 0.75 (0.50 to 1.12), p = 0.162 | 1.01 (0.64 to 1.60), p = 0.965 | 1.63 (1.02 to 2.60), p = 0.042 | 2.03 (1.22 to 3.38), p = 0.006 | 2.98 (1.70 to 5.23), p<0.001 |
| Model 2 | | Ref | 0.73 (0.49 to 1.1), p = 0.132 | 1.00 (0.63 to 1.58), p = 1 | 1.59 (0.99 to 2.54), p = 0.053 | 2.01 (1.21 to 3.33), p = 0.007 | 2.93 (1.67 to 5.13), p<0.001 |
| Model 3 | | Ref | 0.79 (0.53 to 1.19), p = 0.264 | 1.18 (0.74 to 1.88), p = 0.489 | 1.70 (1.05 to 2.75), p = 0.03 | 2.31 (1.36 to 3.90), p = 0.002 | 3.73 (2.04 to 6.84), p<0.001 |
| **In-hospital mortality, number (%)** | 183 (2.62%) | 27 (1.1%) | 49 (2.0%) | 27 (3.0%) | 41 (7.3%) | 27 (8.7%) | 12 (6.8%) |
| Odds ratio (95% CI), p-value | | | | | | | |
| Unadjusted model | | Ref | 1.86 (1.16 to 2.99), p = 0.01 | 2.92 (1.70 to 5.01), p<0.001 | 7.35 (4.48 to 12.06), p<0.001 | 8.93 (5.16 to 15.43), p<0.001 | 6.82 (3.39 to 13.71), p<0.001 |
| Model 1 | | Ref | 0.68 (0.40 to 1.16), p = 0.162 | 0.70 (0.37 to 1.32), p = 0.268 | 1.68 (0.92 to 3.06), p = 0.092 | 2.01 (1.05 to 3.83), p = 0.035 | 2.12 (0.99 to 4.53), p = 0.054 |
| Model 2 | | Ref | 0.67 (0.39 to 1.14), p = 0.138 | 0.69 (0.37 to 1.30), p = 0.254 | 1.64 (0.90 to 2.99), p = 0.107 | 1.98 (1.04 to 3.78), p = 0.037 | 2.07 (0.97 to 4.43), p = 0.06 |
| Model 3 | | Ref | 0.75 (0.44 to 1.27), p = 0.287 | 0.80 (0.42 to 1.50), p = 0.479 | 1.69 (0.92 to 3.10), p = 0.088 | 2.11 (1.09 to 4.08), p = 0.026 | 2.53 (1.13 to 5.64), p = 0.024 |
| **ICU admission, number (%)** | 441 (6.32%) | 98 (3.9%) | 146 (5.8%) | 81 (9.1%) | 51 (9.1%) | 34 (11.0%) | 31 (17.6%) |
| Odds ratio (95% CI), p-value | | | | | | | |
| Unadjusted model | | Ref | 1.55 (1.19 to 2.01), p = 0.001 | 2.50 (1.84 to 3.39), p<0.001 | 2.50 (1.76 to 3.55), p<0.001 | 3.09 (2.05 to 4.65), p<0.001 | 5.34 (3.45 to 8.26), p<0.001 |
| Model 1 | | Ref | 1.47 (1.10 to 1.98), p = 0.01 | 2.33 (1.61 to 3.36), p<0.001 | 2.32 (1.54 to 3.49), p<0.001 | 2.86 (1.80 to 4.55), p<0.001 | 5.05 (3.17 to 8.04), p<0.001 |
| Model 2 | | Ref | 1.45 (1.08 to 1.95), p = 0.014 | 2.31 (1.60 to 3.33), p<0.001 | 2.28 (1.52 to 3.44), p<0.001 | 2.84 (1.79 to 4.50), p<0.001 | 4.97 (3.12 to 7.91), p<0.001 |
| Model 3 | | Ref | 1.43 (1.06 to 1.93), p = 0.02 | 2.11 (1.45 to 3.08), p<0.001 | 2.10 (1.38 to 3.20), p = 0.001 | 2.55 (1.58 to 4.13), p<0.001 | 4.84 (2.96 to 7.90), p<0.001 |
| **Rehospitalisation, number (%)** | 581 (8.32%) | 167 (6.56%) | 223 (8.91%) | 89 (10.01%) | 51 (9.09%) | 29 (9.39%) | 22 (12.50%) |
| Odds ratio (95% CI), p-value | | | | | | | |
| Unadjusted model | | Ref | 1.39 (1.13 to 1.71), p = 0.002 | 1.58 (1.21 to 2.07), p = 0.001 | 1.42 (1.03 to 1.98), p = 0.035 | 1.47 (0.98 to 2.23), p = 0.066 | 2.03 (1.27 to 3.26), p = 0.003 |
| Model 1 | | Ref | 1.28 (1.01 to 1.63), p = 0.043 | 1.40 (1.01 to 1.93), p = 0.041 | 1.25 (0.86 to 1.82), p = 0.238 | 1.29 (0.82 to 2.04), p = 0.266 | 1.85 (1.13 to 3.02), p = 0.015 |
| Model 2 | | Ref | 1.27 (1.00 to 1.61), p = 0.052 | 1.40 (1.01 to 1.93), p = 0.042 | 1.24 (0.86 to 1.81), p = 0.254 | 1.29 (0.82 to 2.03), p = 0.273 | 1.83 (1.12 to 2.99), p = 0.016 |
| Model 3 | | Ref | 1.25 (0.98 to 1.59), p = 0.067 | 1.35 (0.98 to 1.87), p = 0.07 | 1.14 (0.78 to 1.67), p = 0.49 | 1.17 (0.74 to 1.85), p = 0.508 | 1.51 (0.91 to 2.51), p = 0.114 |
| **LOS (days), mean (SD)** | 4.39 (5.82) | 2.78 (5.13) | 4.53 (5.87) | 5.82 (5.82) | 6.24 (6.46) | 7.06 (5.20) | 7.39 (6.26) |
| Unadjusted model | | Ref | 1.76 (1.41 to 2.11), p<0.001 | 3.05 (2.57 to 3.53), p<0.001 | 3.47 (2.89 to 4.04), p<0.001 | 4.29 (3.54 to 5.04), p<0.001 | 4.62 (3.67 to 5.56), p<0.001 |

*(Continued)*

**Table 2.** (Continued)

| | Total cohort | eGFR (ml/min/1.73m$^2$) | | | | | |
|---|---|---|---|---|---|---|---|
| | | >90 | 60–89 | 45–59 | 30–44 | 15–29 | <15 |
| Model 1 | | Ref | 0.60 (0.20 to 1.00), p = 0.003 | 1.32 (0.77 to 1.88), p<0.001 | 1.73 (1.08 to 2.37), p<0.001 | 2.49 (1.68 to 3.29), p<0.001 | 3.40 (2.44 to 4.36), p<0.001 |
| Model 2 | | Ref | 0.58 (0.18 to 0.98), p = 0.004 | 1.34 (0.78 to 1.89), p<0.001 | 1.72 (1.07 to 2.36), p<0.001 | 2.50 (1.70 to 3.31), p<0.001 | 3.37 (2.41 to 4.33), p<0.001 |
| Model 3 | | Ref | 0.55 (0.16 to 0.94), p = 0.005 | 1.09 (0.55 to 1.64), p<0.001 | 1.27 (0.64 to 1.91), p<0.001 | 1.99 (1.20 to 2.78), p<0.001 | 3.16 (2.21 to 4.11), p<0.001 |

95% Confidence interval, eGFR estimated glomerular filtration rate, ICU intensive care unit, LOS length of stay, OR Odds ratio, models were stepwise adjusted for age (model 1), age, and gender (model 2), age, gender, main diagnosis, and comorbidities (model 3)

*Since LOS is a continuous variable regression results represent a regression coefficient (95% confidence interval)

8.80 (95% CI 5.27 to 14.68, p<0.001) significantly higher for the eGFR group <15ml/min/1.73m$^2$.

After adjustment for important confounders (such as age, gender, main diagnosis, and comorbidities) the groups with an eGFR below 45 ml/min/1.73m$^2$ remained significantly associated with 30-day mortality (Table 2). Using multivariate regression the adjusted OR was 1.70 (95% CI, 1.05 to 2.75, p = 0.03) for 30–44 ml/min/1.73m$^2$, 2.31 (95% CI 1.36 to 3.90, p = 0.002) for 15-29ml/min/1.73m$^2$ and 3.73 (95% CI 2.04 to 6.84, p<0.001) for the lowest group (<15ml/min/1.73m$^2$), as compared to the highest eGFR group.

The in-hospital mortality with eGFR <15 ml/min/1.73m$^2$ was lower than in the group with a eGFR of 15–30 ml/min/1.73m$^2$, with a relative small number of patients and events in both groups.

Each decrease in eGFR of 10 ml/min/1.73m$^2$ was associated with a 15% increased risk of 30-day mortality (OR 1.15, 95% CI 1.09 to 1.22, p<0.001) (Table 3).When adjusted for the selected biomarkers Hemoglobin (Hb), C-reactive Protein (CRP), and Copeptin individually, eGFR remained significantly associated with 30-day mortality. When adjusted for all biomarkers at once the association with eGFR was no longer apparent (S2 Table).

**Table 3. Univariate and multivariate logistic regression analyses according to continuous eGFR values.**

| | Unadjusted model | Model 1 | Model 2 | Model 3 |
|---|---|---|---|---|
| | OR (95% CI), p-value | | | |
| **30-day mortality** | | | | |
| eGFR per decrease 10ml/min/1.73m$^2$ | 1.28 (1.24 to 1.33), p<0.001 | 1.15 (1.09 to 1.20), p<0.001 | 1.14 (1.09 to 1.20), p<0.001 | 1.15 (1.09 to 1.22), p<0.001 |
| **In-hospital mortality** | | | | |
| eGFR per decrease 10ml/min/1.73m$^2$ | 1.29 (1.23 to 1.36), p<0.001 | 1.16 (1.09 to 1.24), p<0.001 | 1.16 (1.08 to 1.23), p<0.001 | 1.16 (1.08 to 1.24), p<0.001 |
| **ICU admission** | | | | |
| eGFR per decrease 10ml/min/1.73m$^2$ | 1.16 (1.13 to 1.20), p<0.001 | 1.16 (1.12 to 1.21), p<0.001 | 1.16 (1.12 to 1.21), p<0.001 | 1.15 (1.10 to 1.20), p<0.001 |
| **Rehospitalisation** | | | | |
| eGFR per decrease 10ml/min/1.73m$^2$ | 1.06 (1.03 to 1.09), p<0.001 | 1.05 (1.01 to 1.09), p = 0.015 | 1.05 (1.01 to 1.09), p = 0.018 | 1.03 (0.99 to 1.07), p = 0.132 |
| **LOS** * | | | | |
| eGFR per decrease 10ml/min/1.73m$^2$ | 0.5 (0.45 to 0.55), p<0.001 | 0.30 (0.24 to 0.37), p<0.001 | 0.30 (0.23 to 0.36), p<0.001 | 0.25 (0.19 to 0.32), p<0.001 |

95% CI 95% Confidence interval, eGFR estimated glomerular filtration rate, ICU intensive care unit, LOS length of stay, OR Odds ratio, Models were stepwise adjusted for age (model 1), age, and gender (model 2), age, gender, main diagnosis, and comorbidities (model 3)

*Since LOS is a continuous variable regression results represent a regression coefficient (95% confidence interval)

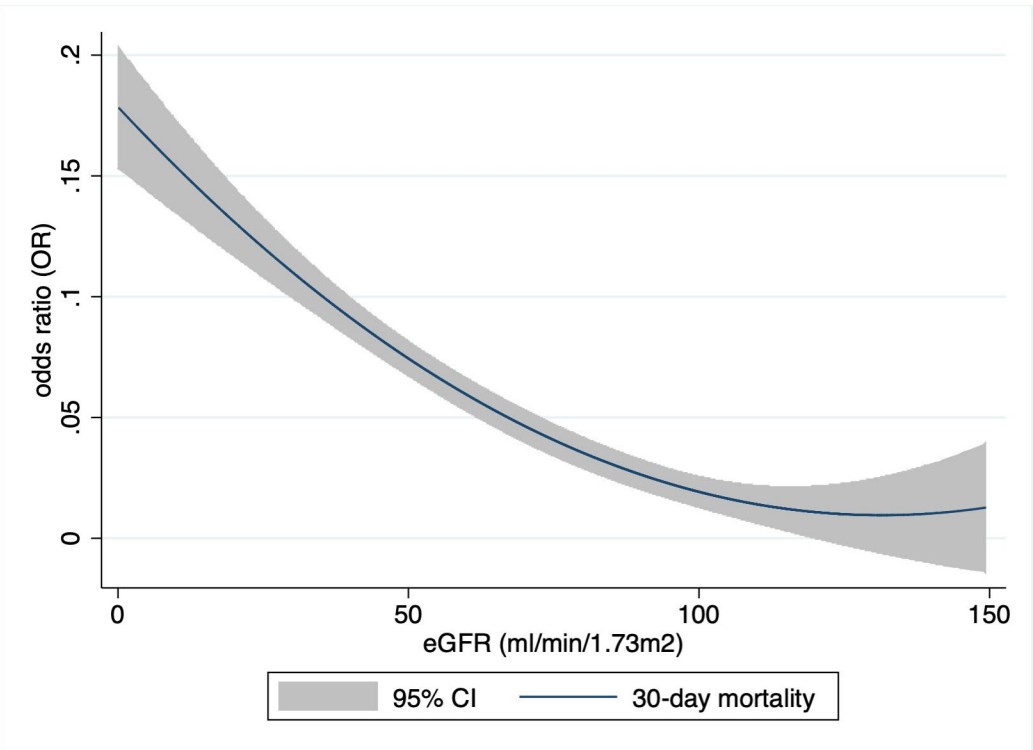

**Fig 1. Hazard ratio of 30-day mortality as a function of eGFR.** (x-axis, eGFR (ml/min/1.73m$^2$), y-axis, odd ratio for 30-day mortality, CI, confidence interval).

There was a 'U'-shaped association between eGFR and 30-day mortality, with a nadir at approximately 130 ml/min/1.73m$^2$ (**Fig 1**). The curve was relatively flat between 100 and 160 ml/min/1.73 m$^2$, but increased sharply at lower and higher levels of eGFR (with the higher levels being outside the physiologic meaningful area. These are not illustrated in the figure). Kaplan-Meier curves show a stepwise increase in mortality in the different eGFR classes (**Fig 2**).

As secondary endpoints the in-hospital mortality, ICU admission, readmission and length of hospital stay were further evaluated.

A decrease in eGFR was significantly associated with in-hospital mortality, ICU-admission, and with a prolonged length of stay (**Table 2,**). After adjustment for confounders we found a significantly higher in-hospital mortality in patients with an eGFR <30 ml/min/1.73 m$^2$, as compared to the reference group.

In contrast, even mild impairment in renal function (<90 ml/min/1.73 m$^2$) was significantly associated with an increased risk for ICU admission and longer hospital stay (**Table 2**).

For every drop of 10 mL/min/1.73 m$^2$ the adjusted OR was 1.15 (95% CI, 1.10 to 1.2, p<0.001), for ICU admission and 1.16 (95% CI1.08 to 1.24, p<0.001) for in-hospital mortality. LOS increased by 0.25 days for every drop of 10 mL/min/1.73 m$^2$ (95% CI 0.19 to 0.32, p<0.001) in the adjusted model (**Table 3**).

There was no significant association between a decrease in eGFR and hospital readmission rate.

## Discriminative performance of eGFR

The receiver operating characteristic (ROC) curve showed acceptable discrimination by eGFR with an AUC of 0.71 (95% CI 0.68 to 0.73) for 30-day mortality and an AUC of 0.71 (95% CI 0.68 to 0.75) for in-hospital mortality in the total cohort (**Table 4**).

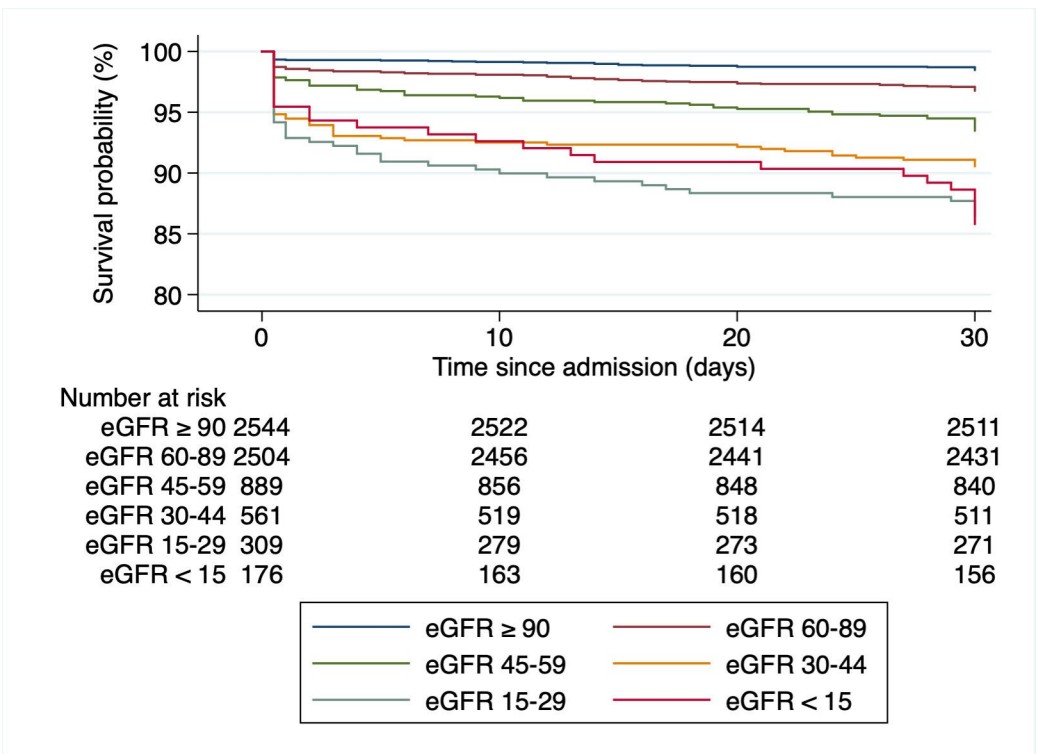

**Fig 2. Kaplan-Meier survival estimates by eGFR groups.** x-axis, day since admission, y-axis, Proportion of patients alive, eGFR, estimated glomerular filtration rate (ml/min/1.73m²).

## Discussion

This study illustrates that eGFR, (using the CKD-EPI equation) from a single serum-creatine measurement, was significantly associated with adverse clinical outcomes, particularly reflected by the 30-day mortality, as well as with in-hospital mortality, ICU admission, and LOS. Decreased eGFR might therefore serve as a general risk marker as reflected in this largest cohort of medical emergency patients studied so far.

The data is consistent with previous studies which described an association between poor outcome and decreased renal function in particular disease entities. These include patients with myocardial infarction [25, 26], heart failure [9, 10, 27], those undergoing percutaneous cardiovascular interventions [28], coronary artery bypass graft surgery [11, 29], and stroke [30, 31]. Our approach was to evaluate the eGFR as a general risk marker irrespective of the final diagnosis for daily routine in emergency care.

**Table 4. Discriminative performance of eGFR and serum creatinine for the prediction of the different outcomes.**

|  | eGFR | Creatinine |
|---|---|---|
|  | AUC | |
| 30-day mortality | 0.71 (0.68 to 0.73) | 0.65 (0.61 to 0.68) |
| In-hospital mortality | 0.71 (0.68 to 0.75) | 0.66 (0.61 to 0.70) |
| ICU admission | 0.63 (0.60 to 0.65) | 0.61 (0.58 to 0.64) |

AUC area under the receiver operating curve, eGFR estimated glomerular filtration rate, ICU intensive care unit.

It is currently not clear whether poor renal function at time of presentation is just a marker for sicker patients or whether there is a causal relationship with the increased mortality. Several factors might be involved in the higher mortality such as electrolyte disturbances, metabolic acidosis, volume overload, anemia and the negative impact of uremic compounds (resulting in e.g. vasculopathy or increased risk for infection).

We would suggest that in patients with mildly reduced renal function the eGFR reduction might be a reflection of a sicker patient population, whereas in severe renal failure the complexity of the treatment will increase significantly, and the causality hypothesis might be true.

The data illustrates a significantly higher mortality risk for the eGFR 30–44 ml/min/1.73m$^2$ group as compared to the group with an eGFR 45–60 ml/min/1.73m$^2$.This supports the decision of the KDIGO to subdivide stage 3 in chronic kidney disease into categories a and b (as also reflected by the risk in the general population) [24].

Patients with an eGFR 15–29 ml/min/1.73m$^2$ had a worse in-hospital mortality as compared with patients with a eGFR below 15 ml/min/1.73m$^2$ (Table 3, Fig 2). This is a surprising finding without a clear explanation. The relatively small patient groups might have led to this result. Both the 30-day mortality and ICU admission rate are higher in patients with an eGFR 15 ml/min/1.73m$^2$ which indicates that this group is actually sicker than the afore mentioned. In-hospital mortality is influenced by hospital transfer policies and therefore the outcome at defined time points (e.g. 30, 60, 90 days) might be the more robust parameter [32, 33]. Our data illustrates a 'U'-shaped relationship between eGFR and all-cause mortality, with increased risk among those with both low and very high values of eGFR, respectivly. This is in line with the findings of Shlipak et al. [34]. They showed that the association between quintiles of creatinine and all-cause mortality appeared to be J-shaped among 4,637 participants in the Cardiovascular Health Study. An elevated risk of cardiovascular events was also found among patients with atherosclerotic cardiovascular disease with an eGFR >125 ml/min/1.73 m$^2$ by Inrig et al. [35]. The eGFR might be elevated due to hyperfiltration in early diabetic nephropathy, diet (e.g. dietary supplements, vegetarians), medication use and rapidly changing kidney function [30]. Patients with abnormal muscle mass might also be included (e.g. amputation, muscular disease, chronically ill).

To the best of our knowledge, this is the largest study using eGFR calculated with the CKD-EPI formula in patients presenting to the medical ED. There has been intensive discussion about which formula might be most suitable in specific situations. For example, Moreno et al. found that the Cockcroft-Gault formula was a slightly better predictor of mortality in acute heart failure patients [20]. The Cockcroft-Gault formula contains the body weight of the patient, a variable that might provide additional information about patients' constitution or nutritional state and might have influenced the shape of our eGFR vs. mortality curve. Unfortunately, as data on body weight were not available in our cohort, we were not able to compare these two.

Several limitations of our study need to be discussed. First, this is a secondary analysis of a previous observational study. We are therefore limited to the parameters collected in the primary study. Important factors which are not available are the cause of death, the course of renal function (i.e. acute kidney injury versus chronic kidney disease), and the number of patients on dialysis.

The single measurement of serum creatinine at hospital admission cannot discriminate between acute kidney injury, acute kidney disease and chronic kidney disease. Moreover, eGFR levels might change over the course of a patients stay. It would have been an interesting question whether the risk differs in acute versus chronic renal impairment, and also whether improvement of renal function during the hospital stay would be associated with better outcome.

As dialysis patients might bias the group with an eGFR <15 ml/min/1.73m$^2$, an analysis was performed without these patients potentially on hemodialysis. The eGFR remained significantly associated with 30-day mortality, both as a continuous variable, and for patients with an eGFR 30–45 ml/min/1.73m$^2$ and 15–30 ml/min/1.73m$^2$ (S3 Table). Although, the number of dialysis patients in this analysis might be higher than in the general population, the absolute number is expected to be low (but still an important source of bias). Whether a risk stratification with eGFR would lead to better patient care and to a lowering of adverse outcome in this unselected population cannot be concluded. Causality cannot be implied by our data. We hope to create further hypotheses and result in studies with detailed analysis of the course of renal failure.

## Conclusion

Decreased renal function reflected by a reduction in eGFR (CKD-EPI), at time of ED admission was associated with increased risk of 30-day mortality, ICU admission, and length hospital stay. The eGFR may serve as a useful tool for risk stratification in the medical emergency department.

## Supporting information

**S1 Table. Baseline characteristics of the total cohort by centers.**
(DOCX)

**S2 Table. Associations of eGFR with adverse clinical outcome in an univariate model and after adjustment for laboratory results.**
(DOCX)

**S3 Table. Associations of eGFR with adverse clinical outcome in univariate and multivariate models without patients with a eGFR<15 ml/min/1.73m$^2$.**
(DOCX)

## Acknowledgments

All authors made significant intellectual contributions to this study with respect to conception, design, and have taken an active part in acquisition, analysis and interpretation of data. L.H, A. E. S.S. and P.S. conducted statistical analyses and drafted the first manuscript. All the authors have accepted responsibility for the entire content of this submitted manuscript and approved submission.

## Author Contributions

**Conceptualization:** Andreas Eckart, Sebastian Haubitz, Beat Mueller, Philipp Schuetz, Stephan Segerer.

**Data curation:** Andreas Eckart, Sebastian Haubitz, Beat Mueller, Philipp Schuetz.

**Formal analysis:** Laurent Haas, Andreas Eckart, Sebastian Haubitz, Beat Mueller, Philipp Schuetz.

**Funding acquisition:** Beat Mueller, Philipp Schuetz.

**Investigation:** Laurent Haas, Beat Mueller, Philipp Schuetz, Stephan Segerer.

**Methodology:** Beat Mueller, Philipp Schuetz.

**Project administration:** Andreas Eckart, Sebastian Haubitz, Beat Mueller, Philipp Schuetz.

**Resources:** Philipp Schuetz, Stephan Segerer.

**Software:** Philipp Schuetz.

**Supervision:** Philipp Schuetz, Stephan Segerer.

**Writing – original draft:** Laurent Haas.

**Writing – review & editing:** Andreas Eckart, Sebastian Haubitz, Beat Mueller, Philipp Schuetz, Stephan Segerer.

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
