## [Decision Letter · Decision Letter 0]

26 Sep 2019

PONE-D-19-23285

Estimated glomerular filtration rate predicts 30-day mortality in medical emergency departments: results of a prospective multi-national observational study

PLOS ONE

Dear Mr Haas,

Thank you for submitting your manuscript to PLOS ONE. After careful consideration, we feel that it has merit but does not fully meet PLOS ONE’s publication criteria as it currently stands. Therefore, we invite you to submit a revised version of the manuscript that addresses the points raised during the review process.

We would appreciate receiving your revised manuscript by Nov 10 2019 11:59PM. To enhance the reproducibility of your results, we recommend that if applicable you deposit your laboratory protocols in protocols.io, where a protocol can be assigned its own identifier (DOI) such that it can be cited independently in the future. For instructions see: http://journals.plos.org/plosone/s/submission-guidelines#loc-laboratory-protocols

We look forward to receiving your revised manuscript.

Kind regards,

Tatsuo Shimosawa, M.D., Ph.D.

Academic Editor

PLOS ONE

Journal Requirements:

1. Thank you for including your competing interests statement; "none"

3. Please amend your authorship list in your manuscript file to include author Beat Müller

4. Please amend the manuscript submission data (via Edit Submission) to include author Beat Mueller

Reviewers' comments:

Reviewer's Responses to Questions

**Comments to the Author**

1. Is the manuscript technically sound, and do the data support the conclusions?

Reviewer #1: Partly

Reviewer #2: Partly

2. Has the statistical analysis been performed appropriately and rigorously? 

Reviewer #1: I Don't Know

Reviewer #2: Yes

3. Have the authors made all data underlying the findings in their manuscript fully available?

Reviewer #1: Yes

Reviewer #2: Yes

4. Is the manuscript presented in an intelligible fashion and written in standard English?

Reviewer #1: Yes

Reviewer #2: Yes

5. Review Comments to the Author

Reviewer #1: The authors studied the outcome of patients in the ED according to eGFR by the CKD-EPI formula.

1) Primary and secondary outcomes were adjusted for age, gender, main diagnosis, and comorbidities. Why were laboratory data not adjusted for? It seems that those with lower eGFR had higher CRP, copeptin and lower Hb which are probably related to the outcomes.

2) The proportion and basic characteristics of patients recruited from the three tertiary care centers should be mentioned (perhaps as supplementary data).

3) Limitation of not having data on race should be mentioned since this will incorrectly calculate eGFR and race is also probably an important confounding factor of the outcome. Is data on race really not available at least for the cohort from the US?

4) I am not really convinced that this study states a novel insight because serum Cr or eGFR has already been shown to be a predictive factor in chronic and acute settings. Mixing the chronic and acute setting will probably yield thee same results. Can you be more specific on how the result or method of this study is new?

Reviewer #2: Haas and colleagues evaluated the predictive performance of eGFR measured at ED for 30 day mortality. They found a step wise increase of mortality with reduction of eGFR, so eGFR seems to be a good predictor.

1) Were end-stage renal disease patients treated by maintenance dialysis included? If so, their eGFR should be different between before and after dialysis. This will cause a serious bias.

2) Cat 4 (eGFR 15-29) showed worse survival curve just before day 30 compared with Cat 5 (eGFR<15) (figure 2). The reason for this should be investigated. Did Cat 5 include many dialysis patients??

3) It is a little bit surprising that most (72.8%) pts were treated as inpatients in this ED cohort. Can the severity of the enrolled patients be evaluated? Can qSOFA score be caluculated?

4) As mentioned in the limitation section, this study did not distinguish AKI from CKD. However, because 72% of the enrolled patients admitted, subanalysis regarding AKI, CKD, acute-on-chronic can be performed when limited to the inpatients. This analysis will provide more precise role of single measurement of eGFR at ED.

6. PLOS authors have the option to publish the peer review history of their article (what does this mean?). If published, this will include your full peer review and any attached files.

Reviewer #1: Yes: Hiroo Kawarazaki

Reviewer #2: Yes: Kento Doi

---

## [Author Response · Author response to Decision Letter 0]

6 Nov 2019

Mr. Tatsuo Shimosawa, M.D., Ph.D.

Basel, the 5th of November, 2019 

Dear Doctor Shimosawa,

Thank you, for the opportunity to submit a revised manuscript of our study “Estimated glomerular filtration rate predicts 30-day mortality in medical emergency departments: results of a prospective multi-national observational study.” ID: PONE-D-19-23285) for publication in PLOS ONE. We appreciate the time invested in the manuscript by the editorial team and particularly the referees. The suggestions helped us to significantly improve the manuscript. The response to the question of the editorial team and the referees are listed in the point to point answer below.

We hope that the manuscript has reached publication quality for PLOS ONE.

Best regards

Laurent Haas, for all the co-authors

 

To Reviewer #1:

Dear Doctor Kawarazaki,

Thank you very much for the significant time, which you put into the evaluation of our manuscript. We appreciate your constructive comments, which helped us to improve the manuscript. We have incorporated your comments as follows.

Best regards

Laurent Haas.

1) Primary and secondary outcomes were adjusted for age, gender, main diagnosis, and comorbidities. Why were laboratory data not adjusted for? It seems that those with lower eGFR had higher CRP, copeptin and lower Hb which are probably related to the outcomes.

According to your suggestion we performed another analysis adjusting for the laboratory markers and added a table (Table A2) in the supplementary material.

When adjusted for selected variables (Hb, CRP, copeptin), the statistical significance of our main explanatory variable of interest, eGFR, becomes less strong. This suggests that copeptin and eGFR capture some of the same variation with respect to our outcome variable. 

2) The proportion and basic characteristics of patients recruited from the three tertiary care centers should be mentioned (perhaps as supplementary data).

We have added the number of patients per country/center in the results section of the main manuscript (revised manuscript Page 8 paragraph 1).

Moreover, we have added patient characteristic stratified by center as supplementary data (Supplementary Table A4).

3) Limitation of not having data on race should be mentioned since this will incorrectly calculate eGFR and race is also probably an important confounding factor of the outcome. Is data on race really not available at least for the cohort from the US?

Race has of course a significant influence on eGFR, but this information was unfortunately not entered in the utilized database. We regret this limitation with regard to our data. We have added a note of caution and further discussed that in section of the limitation of the study (revised manuscript page 18 paragraph 2).

4) I am not really convinced that this study states a novel insight because serum Cr or eGFR has already been shown to be a predictive factor in chronic and acute settings. Mixing the chronic and acute setting will probably yield thee same results. Can you be more specific on how the result or method of this study is new?

Thank you for this important comment. As outlined in the introduction of our manuscript multiple studies assessing the predictive power of SCr and eGFR in a chronic setting. We are not aware of any larger studies of emergency patients. Given the necessity for a fast and easy (cost-efficient) biomarker for risk stratification in an emergency setting, we believe that it is worthwhile to assess its aptitude with respect to this setting. But it is correct that for the diseases already studied, our data is confirmative in nature. This was discussed in more detail (page 4 paragraph 3). 

 

Reviewer #2: 

Dear Doctor Kent Doi,

We appreciate your time and effort which you put into the evaluation of our manuscript. We think that these helped us to significantly improve the revised manuscripted. We have incorporated your comments as follows.

Best regards

Laurent Haas.

1) Were end-stage renal disease patients treated by maintenance dialysis included? If so, their eGFR should be different between before and after dialysis. This will cause a serious bias.

As rightly suggested chronic hemodialysis HD patients have highly different creatinine values before and after dialysis, and therefore a high variance in eGFR. Hemodialysis patients were included in our study although this variable was not surveyed, so our models could not be adjusted for this circumstance. The exact proportion of this patients in the cohort is unknown. We assume that although it is higher than in the general population, the absolute quantity should be relatively low. Dialysis patients have an increased mortality rate. This should also be reflected in the data with high baseline creatinine (even after dialysis) compared to the general population. To address your comment we have added a table (Table A3) with our calculations excluding patients with Cat 5 patients (as this group contains patients on dialysis) in the supplementary materials. These results are described in in the revised manuscript on page 16 paragraph 4). The results remain robust after excluding Cat 5 patients.

2) Cat 4 (eGFR 15-29) showed worse survival curve just before day 30 compared with Cat 5 (eGFR<15) (Figure 2). The reason for this should be investigated. Did Cat 5 include many dialysis patients?

This is indeed a curious finding, as the Cat. 4 patients seem to have a higher in hospital mortality but a lower 30-day mortality. We added the following paragraph in our manuscript (Page 16, Paragraph 3) Cat.5. patients have a higher ICU admission rate. These patients could be perceived as “sicker” and more short-term medical resources might be attributed (as reflected by ICU admission) to them, influencing discharge and transfer practices. Experts are discussing in-hospital vs 30-day mortality for outcome prediction/quality assessment and 30-day mortality may be a more valid measure, since it is a fixed point in time. In-hospital mortality is primary influenced by hospital transfer policies and long term (30, 60, 90 days) is mainly determined by diagnosis [1, 2]. Mortality seems to “catch up” with the Cat.5 patients on day 30.

The number of hemodialysis patients has been addressed under the first question.

3) It is a little bit surprising that most (72.8%) pts were treated as inpatients in this ED cohort. Can the severity of the enrolled patients be evaluated? Can qSOFA score be calculated?

We now provide frequency of inpatient care in the newly added Table A2 in the supplementary data (please see also our answer to question 2 of Reviewer #1). As you can see, there was a broad difference in frequency of inpatient care in the participating hospitals. This can be explained by the different health care systems studied. In the US-hospital and in the Swiss hospital medical patients are admitted through the emergency department for inpatient care, and patients with high probability of outpatient care are seen in a separate department (outpatient emergency service). In the French hospital, all of the patients are seen in the same department, which explains the distinct difference in inpatient treatment. 

Unfortunately, data for respiratory rate is limited. That’s why we are not able to calculate SOFA score in a meaningful number of patients. 

4) As mentioned in the limitation section, this study did not distinguish AKI from CKD. However, because 72% of the enrolled patients admitted, subanalysis regarding AKI, CKD, acute-on-chronic can be performed when limited to the inpatients. This analysis will provide more precise role of single measurement of eGFR at ED.

Indeed, most patients were treated as inpatients. In the database, there is only a one-time creatinine measurement available to us. During hospitalization creatinine monitoring was certainly performed but it was not integrated in the database, as the initial study design was to analyze the predictive power of novel biomarkers during triage in the emergency department. A follow up creatinine would have been interesting to differentiate between AKI and CKD. This would have provided additional information to the data and value to the paper. 

We are aware of the limitations of the data; however, we believe that our data provide interesting information due to its large sample size and real-life cohort. It is a good reflection of the diverse patient population encountered in daily practice. We have emphasized this limitation in the revised manuscript (page 17 paragraph 3).

References:

1. van Rijn M, Buurman BM, Macneil Vroomen JL, Suijker JJ, ter Riet G, Moll van Charante EP, et al. Changes in the in-hospital mortality and 30-day post-discharge mortality in acutely admitted older patients: retrospective observational study. Age and Ageing. 2016;45(1):41-7. doi: 10.1093/ageing/afv165.

2. Vasilevskis EE, Kuzniewicz MW, Dean ML, Clay T, Vittinghoff E, Rennie DJ, et al. Relationship between discharge practices and intensive care unit in-hospital mortality performance: evidence of a discharge bias. Med Care. 2009;47(7):803-12. Epub 2009/06/19. doi: 10.1097/MLR.0b013e3181a39454. PubMed PMID: 19536006.

---

## [Decision Letter · Decision Letter 1]

29 Nov 2019

PONE-D-19-23285R1

Estimated glomerular filtration rate predicts 30-day mortality in medical emergency departments: results of a prospective multi-national observational study

PLOS ONE

Dear Mr Haas,

Thank you for submitting your manuscript to PLOS ONE. After careful consideration, we feel that it has merit but does not fully meet PLOS ONE’s publication criteria as it currently stands. Therefore, we invite you to submit a revised version of the manuscript that addresses the points raised during the review process.

The Policy of PLoS One request authors to provide all the data available.

We would appreciate receiving your revised manuscript by Jan 13 2020 11:59PM. To enhance the reproducibility of your results, we recommend that if applicable you deposit your laboratory protocols in protocols.io, where a protocol can be assigned its own identifier (DOI) such that it can be cited independently in the future. For instructions see: http://journals.plos.org/plosone/s/submission-guidelines#loc-laboratory-protocols

We look forward to receiving your revised manuscript.

Kind regards,

Tatsuo Shimosawa, M.D., Ph.D.

Academic Editor

PLOS ONE

Reviewers' comments:

Reviewer's Responses to Questions

**Comments to the Author**

1. If the authors have adequately addressed your comments raised in a previous round of review and you feel that this manuscript is now acceptable for publication, you may indicate that here to bypass the “Comments to the Author” section, enter your conflict of interest statement in the “Confidential to Editor” section, and submit your "Accept" recommendation.

Reviewer #1: All comments have been addressed

Reviewer #2: (No Response)

2. Is the manuscript technically sound, and do the data support the conclusions?

Reviewer #1: Yes

Reviewer #2: No

3. Has the statistical analysis been performed appropriately and rigorously? 

Reviewer #1: Yes

Reviewer #2: Yes

4. Have the authors made all data underlying the findings in their manuscript fully available?

Reviewer #1: Yes

Reviewer #2: Yes

5. Is the manuscript presented in an intelligible fashion and written in standard English?

Reviewer #1: Yes

Reviewer #2: Yes

6. Review Comments to the Author

Reviewer #1: Just one minor comment. Table A4 has systolic BP and diastolic BP the other way round. All comments have been addressed.

Reviewer #2: Unfortunately, the authors could not add any data because of data availability of their database. That suggests this study has numbers of significant limitations.

7. PLOS authors have the option to publish the peer review history of their article (what does this mean?). If published, this will include your full peer review and any attached files.

Reviewer #1: Yes: Hiroo Kawarazaki

Reviewer #2: No

---

## [Author Response · Author response to Decision Letter 1]

27 Feb 2020

To Reviewer #1:

Dear Doctor Kawarazaki,

Thank you very much for your comment (as addressed below, and your positive evaluation. We really appreciate your time invested.

Best regards

Laurent Haas

1) Just one minor comment. Table A4 has systolic BP and diastolic BP the other way round. All comments have been addressed.

Thank you for your perceptiveness. We have corrected the mistake and also some minor formatting issues in our tables.  

Reviewer #2: 

Dear Doctor Kent Doi,

Thank you for your work on our manuscript. In your comment you pointed to the limitations of the study and particularly the missing information. With this in mind, we returned to your comments of the first review. In this second revision we further addressed your comments as follows.

1) Were end-stage renal disease patients treated by maintenance dialysis included? If so, their eGFR should be different between before and after dialysis. This will cause a serious bias.

The text was extensively revised. Excluding patients with an eGFR below 15 ml/min (Supplemental Material Table A3) did not change the results significantly. The limitation was included in the Abstract and Discussion, respectively.

2) Cat 4 (eGFR 15-29) showed worse survival curve just before day 30 compared with Cat 5 (eGFR<15) (Figure 2). The reason for this should be investigated. Did Cat 5 include many dialysis patients?

This remains as an intriguing finding. The number of patients and the event rate is small in patients with a eGFR<15 ml/min. Only three events would have lead to the same event rate as in the group with a eGFR 15-30 ml/min. Therefore, the group size might be of importance. Furthermore, 30-day mortality might be the better parameter as discussed. 

3) It is a little bit surprising that most (72.8%) pts were treated as inpatients in this ED cohort. Can the severity of the enrolled patients be evaluated? Can qSOFA score be calculated?

Our ED consist of two wards. One is managed by general physicians and foreseen for patients likely to be treated as outpatients. In the other ward, patients likely to be treated as inpatients are assigned to. The study enrolled only patients in the latter ward, which explains the high rate of inpatients. 

qSOFA cannot be calculated as the information is not available. The qSOFA would describe the severity in septic patients. Only, 15% suffered from infectious diseases. Therefore, the gained information would be limited.

4) As mentioned in the limitation section, this study did not distinguish AKI from CKD. However, because 72% of the enrolled patients admitted, subanalysis regarding AKI, CKD, acute-on-chronic can be performed when limited to the inpatients. This analysis will provide more precise role of single measurement of eGFR at ED.

Based on the helpful suggestions of the reviewers, the text was extensively revised and focused on the answers which can be given with the current dataset.

---

## [Editor Report · Decision Letter 2]

16 Mar 2020

Estimated glomerular filtration rate predicts 30-day mortality in medical emergency departments: results of a prospective multi-national observational study

PONE-D-19-23285R2

Dear Dr. Haas,

We are pleased to inform you that your manuscript has been judged scientifically suitable for publication and will be formally accepted for publication once it complies with all outstanding technical requirements.

With kind regards,

Tatsuo Shimosawa, M.D., Ph.D.

Academic Editor

PLOS ONE
---

## [Editor Report · Acceptance letter]

23 Mar 2020

PONE-D-19-23285R2 

Estimated glomerular filtration rate predicts 30-day mortality in medical emergency departments: Results of a prospective multi-national observational study 

Dear Dr. Haas:

I am pleased to inform you that your manuscript has been deemed suitable for publication in PLOS ONE. Congratulations! Your manuscript is now with our production department. 

With kind regards,

on behalf of

Prof. Tatsuo Shimosawa 

Academic Editor

PLOS ONE